# Rational Design of Resveratrol O-methyltransferase for the Production of Pinostilbene

**DOI:** 10.3390/ijms22094345

**Published:** 2021-04-21

**Authors:** Daniela P. Herrera, Andrea M. Chánique, Ascensión Martínez-Márquez, Roque Bru-Martínez, Robert Kourist, Loreto P. Parra, Andreas Schüller

**Affiliations:** 1Department of Chemical and Bioprocesses Engineering, School of Engineering, Pontificia Universidad Católica de Chile, Vicuña Mackenna 4860, Santiago 7820244, Chile; dpherrer@uc.cl (D.P.H.); amchanique@uc.cl (A.M.C.); 2Institute of Molecular Biotechnology, Graz University of Technology, Petersgasse 14, 8010 Graz, Austria; kourist@tugraz.at; 3Department of Agrochemistry and Biochemistry, Faculty of Science and Multidisciplinary Institute for Environmental Studies “Ramon Margalef”, University of Alicante, 03690 Alicante, Spain; asun.martinez@gcloud.ua.es (A.M.-M.); roque.bru@gcloud.ua.es (R.B.-M.); 4Institute for Biological and Medical Engineering, Schools of Engineering, Medicine and Biological Sciences, Pontificia Universidad Católica de Chile, Vicuña Mackenna 4860, Santiago 7820244, Chile; 5Department of Molecular Genetics and Microbiology, School of Biological Sciences, Pontificia Universidad Católica de Chile, Av. Libertador Bernardo O’Higgins 340, Santiago 8320000, Chile

**Keywords:** enzyme engineering, O-methyltransferases, pinostilbene, protein models, substrate selectivity, stilbenes

## Abstract

Pinostilbene is a monomethyl ether analog of the well-known nutraceutical resveratrol. Both compounds have health-promoting properties, but the latter undergoes rapid metabolization and has low bioavailability. O-methylation improves the stability and bioavailability of resveratrol. In plants, these reactions are performed by O-methyltransferases (OMTs). Few efficient OMTs that monomethylate resveratrol to yield pinostilbene have been described so far. Here, we report the engineering of a resveratrol OMT from *Vitis vinifera* (VvROMT), which has the highest catalytic efficiency in di-methylating resveratrol to yield pterostilbene. In the absence of a crystal structure, we constructed a three-dimensional protein model of VvROMT and identified four critical binding site residues by applying different in silico approaches. We performed point mutations in these positions generating W20A, F24A, F311A, and F318A variants, which greatly reduced resveratrol’s enzymatic conversion. Then, we rationally designed eight variants through comparison of the binding site residues with other stilbene OMTs. We successfully modified the native substrate selectivity of VvROMT. Variant L117F/F311W showed the highest conversion to pinostilbene, and variant L117F presented an overall increase in enzymatic activity. Our results suggest that VvROMT has potential for the tailor-made production of stilbenes.

## 1. Introduction

Stilbenes are phenolic compounds derived from the secondary metabolism of plants that participate in their constitutive and inducible defense mechanisms [1]. The backbone of their chemical structure corresponds to 1,2-diphenylethylene [1]. Resveratrol (trans-3,5,4′-trihydroxystilbene) is one of the most studied stilbenes due to its health-promoting properties with anticancer [2], antioxidant [3], anti-inflammatory [4,5], cardioprotective [6], and neuroprotective activities [7]. Despite these properties, resveratrol has less than 1% of oral bioavailability in humans [8], an 8–14 min half-life [9,10], and undergoes rapid metabolization due to glucuronidation, sulfation, and hydrogenation reactions of its aliphatic double bond [11]. Whether these metabolic products have biological activity remains unclear [8,12]. The low bioavailability of resveratrol still represents a challenge. It has been shown that protection of resveratrol’s 3 and 5-hydroxyl moieties by O-methylation increases lipophilicity, stability, uptake into human cells, and therefore its bioavailability [13,14].

Pterostilbene (trans-3,5-dimethoxy-4-hydroxystilbene), a dimethylated derivate, has a superior pharmacokinetic profile with a longer elimination half-life of 94–103 min and oral bioavailability of 13.4–16% [13,14], increasing to 59% when a cyclodextrin formulation is used [14]. On the other hand, the less studied pinostilbene (trans-3-methoxy-5,4′-dihydroxystilbene) was recently detected in *Commiphora africana*, a medicinal plant used for cancer treatment [15]. Several studies reported that pinostilbene exerted anticancer [15,16], antioxidant [17], and neuroprotective properties [18].

The developed processes to obtain stilbenes consist of chemical synthesis, direct plant extraction, or production in different biological platforms. These latter bioproduction strategies involve more efficient and environment-friendly conditions. Among them, plant cell cultures [19] and microbial cell systems [20] (principally the species *Saccharomyces cerevisiae* [21] and *Escherichia coli* [22]), are the most widely used. To produce the less common stilbenes it is necessary to integrate an enzyme with the required activity into a biosynthetic pathway. For the methylated stilbenes, the addition of 3/5-resveratrol-O-methyltransferase (OMT) activity would yield either pinostilbene, pterostilbene, or a mixture thereof (Scheme 1).

Plant OMTs comprise three types of enzymes. Type-I OMTs are cation-independent with a molecular weight of 38–43 kDa and act on a diverse group of compounds. Type-II OMTs are cation-dependent, with a molecular weight of 23–27 kDa, and are mostly specific for caffeoyl coenzyme-A esters (CCoAOMTs), and type-III OMTs are principally active on carboxylic acids [23]. Most OMTs used for the bioproduction of methylated stilbenes belong to type-I (OMT-I) including resveratrol OMT from *Vitis vinifera* (VvROMT) [21,24,25], OMT3 from *Sorghum bicolor* (SbOMT3) [21,26], and the promiscuous flavone 3’-OMT from *Arabidopsis thaliana* (AtOMT1) [22].

Few attempts for pinostilbene production have been carried out as most OMT-I described so far catalyze the two-step di-methylation of resveratrol obtaining pterostilbene as the final product. The enzyme SbOMT3 has been used for pinostilbene production [21], albeit with low catalytic efficiency (kcat/Km 1.4 s^−1^M^−1^) [27]. Interestingly, SbOMT3 produces exclusively pterostilbene in plants [27], but mostly pinostilbene when it was recombinantly expressed in *E. coli*, reaching a titer of 34 mg L^−1^, which represents a conversion of 14% to pinostilbene after 60 h [26]. The enzyme AtOMT1 yielded pterostilbene as the principal product (34 mg L^−1^) but also 17.4 mg L^−1^ of pinostilbene when *E. coli* was used as expression system [22]. By contrast, the highly efficient VvROMT produces less than 2 mg L^−1^ of pinostilbene and 35 mg L^−1^ of pterostilbene in *S. cerevisiae* after 72 h [21], as this enzyme rapidly transforms the generated pinostilbene to pterostilbene [28]. Given that VvROMT has the highest catalytic efficiency among the known enzymes that catalyze the di-methylation of resveratrol (5160 s^−1^ M^−1^) [28] yielding pterostilbene, we decided to modify its substrate preference by rational design to obtain a higher proportion of pinostilbene from the enzymatic monomethylation of resveratrol. To our knowledge, there is just one study that attempted to produce pinostilbene with VvROMT. Guan Yuekai and co-workers [29] performed directed evolution on VvROMT to increase pinostilbene production. In a biotransformation assay starting from resveratrol, their best variant, F311W, reached 103.58 mg L^−1^ of pinostilbene after 12 h, which represents a conversion of 45% to pinostilbene, compared with 24.2 mg L^−1^ (11%) of the wildtype enzyme. Despite the improvement in pinostilbene production, they do not mention the yield of pterostilbene, and therefore, the substrate selectivity of these variants is not clear.

In order to apply rational protein engineering techniques, we required structural information of the enzyme and catalytic mechanism [30]. Unfortunately, a crystal structure of VvROMT is not yet available. Here, we report the construction of a three-dimensional protein structure model of VvROMT in a closed and catalytically competent conformation in complex with the co-substrate S-adenosylmethionine (SAM) and resveratrol. Then, applying different in silico approaches, we identified critical residues for VvROMT’s enzymatic activity. Next, we employed rational design to modify the substrate selectivity of VvROMT through comparison of the predicted substrate-binding site residues from our model with other type-I OMTs active on stilbenes. As a result, we obtained variants that preferentially produce the monomethylated resveratrol and that could be potentially integrated in a biosynthetic pathway for pinostilbene production in both metabolically engineered plant cell cultures [19] or microbial systems [21,22].

## 2. Results and Discussion

### 2.1. Phylogenetic Tree of VvROMT and Related OMT-I Enzymes

We generated a phylogram to analyze OMT-I sequences related to VvROMT available in the UniProt/SwissProt database. A total of 70 OMT-I sequences were classified according to their preferred phenylpropanoid substrate and regioselectivity (Figure 1). We focused on enzymes that accept stilbenes to get an insight into their distribution among different OMT-I. Specific and promiscuous OMT-I active on stilbenes clustered separately (Figure 1, Appendix A). VvROMT clustered in a highly diverse group, where the closest related enzymes corresponded to the orcinol OMTs (RhOOMT1-2-4) from *Rosa hybrida*. In fact, RhOOMT4, with 3/5-OH regioselectivity, has been used in grapevine cell cultures to obtain pterostilbene [31] and a homologous orcinol OMT from *Rosa chinensis* was used to identify VvROMT for the first time [28]. The most distantly related enzyme in the VvROMT clade is the specific resveratrol OMT (AcOMT1) from *Acorus calamus*, with 4′-OH regioselectivity for resveratrol [32]. Among the promiscuous stilbene OMT-I (pink stars in Figure 1) we found AtOMT1 and the flavonoid 3’ OMT from *Oryza sativa* (OsOMT1) [33]. Both enzymes generate pterostilbene from resveratrol with 3/5 regioselectivity but are classified as flavonoid OMTs. The pinosylvin OMT1 (PsPMT1) [34] and pinosylvin OMT2 (PsPMT2) [35] from *Pinus silvestris* clustered far from VvROMT. This example of convergent evolution between Scots pine and grapevine has been previously reported [28,35]. These two enzymes are located closer to enzymes active on flavonoids (yellow label) and caffeic acid/5-hydroxyferulic acid (green label). While PsPMT2 is highly specific for the stilbene pinosylvin (trans-3,5-dihydroxystilbene), PsPMT1 has an extended substrate acceptance scope (Appendix A) but both enzymes perform monomethylation reactions with pinosylvin. Considering the substrate preference of type-I OMTs, Wang et al., (1998) reported that (iso)eugenol OMT (CbIEMT1) from *Clarkia breweri* recently evolved from caffeic acid OMT (CbCOMT1) (Figure 1). The former enzyme presents modification in its substrate specificity and regioselectivity [36]. While CbCOMT1 accepts caffeic acid/5-hydroxyferulic acid and has 3/5-OH regioselectivity, CbIEMT1 accepts eugenol/isoeugenol with 4-OH regioselectivity. They found seven residues were responsible for this modification. This case is an example of the flexibility reported for some OMT-I family members. The change of a few residues at the OMT-I substrate-binding site could modify their regioselectivity and substrate specificity. Other rational and semi-rational designs have been performed with CbIEMT1, increasing its substrate scope [37] or generating a highly specific enzyme [38].

This sequence analysis provided important insight about conserved residues for the subsequent structural modelling.

### 2.2. Three-Dimensional Protein Model of VvROMT

We constructed a three-dimensional protein model of VvROMT by comparative protein structure modeling with MODELLER, employing two structural templates, the isoflavone OMT from *Medicago sativa* (Ms7IOMT8; PDB ID:1FP2), and the caffeic acid OMT from *Lolium perenne* (LpCOMT1; PDB ID: 3P9I), with 48.2% and 33.1% sequence identity with VvROMT, respectively. The model represents a homodimer in a closed conformation in complex with the substrate resveratrol and the co-substrate SAM (Figure 2A). Plant type-I OMTs typically consist of a C-terminal SAM binding domain featuring a Rossmann fold and an N-terminal dimerization domain [39]. During dimerization, the two N-terminal domains form a central core structure with the two lateral SAM binding domains attached by a hinge region. The substrate-binding site is located on the central core. In the closed conformation the SAM binding domain is rotated towards the substrate-binding site [39]. The homodimeric state is necessary for the formation of the substrate-binding pocket and, therefore, for the enzymatic activity of OMT-I [40]. In our VvROMT model, the binding residues belonging to the dimeric interphase correspond to W20 and F24, both donated from the sister monomer (Figure 2B). The backbone of W20 possibly stabilizes the 4′-OH of resveratrol, and F24 interacts by stacking with resveratrol’s B-ring.

SAM is stabilized by a hydrogen bond network and van der Waals interactions [40]. The side chain of D243 and the backbone of M244 form hydrogen bonds with the adenine portion of SAM and the hydroxyl ribose ring of SAM interacts with the side chain of D223, a conserved acidic residue in OMT-I [41]. The backbones of G200 and K257 form hydrogen bonds with the SAM’s methionine moiety (Figure 2B).

According to the model, the VvROMT substrate-binding site is comprised of mostly hydrophobic residues. Based on molecular docking results, resveratrol is bound in a catalytically competent conformation with the 3-OH group oriented towards SAM’s sulphonium group (2.96 ± 0.02 Å distance) (Figure 2B). A hydrogen bond is formed between the 3-OH of resveratrol and the side chain of H261 (catalytic residue) and D262. Furthermore, the side chains of F24, F167, F311, and F318 stabilize resveratrol with aromatic interactions, and hydrophobic interactions are observed with residues L117, M121, M171, W258, and M315. A total of 21 residues conform the substrate-binding site within 6 Å of resveratrol (Figure 2B).

To our knowledge, this is the first VvROMT model generated in a closed and catalytically competent conformation. Wang et al. (2014) employed an open conformational state model to identify critical residues of VvROMT [24]. The null enzymatic activity of the variant H261A, constructed by them, corroborated its role as the principal catalytic residue. Generally, methylation in OMT-I involves base-assisted deprotonation of a nucleophilic hydroxyl group [40]. In VvROMT, H261 acts as the base (stabilized by the conserved residue E382) to deprotonate resveratrol’s 3 or 5-hydroxyl group, followed by a nucleophilic attack on the reactive methyl group of SAM to complete the transmethylation reaction.

To guide the rational design of VvROMT variants, additional in silico analyses were performed based on the generated VvROMT model, identifying critical residues at the substrate-binding site level.

### 2.3. Identification of Critical Residues of VvROMT

We analyzed the resveratrol binding site of the VvROMT model with three in silico tools to obtain information about binding residues and their conservation pattern (Figure 3). The buried surface area (BSA) was calculated with our software tool dr_sasa [42]. The hydrophobic residues W20 (15.4 Å2), F24 (19.6 Å2), M121 (15.4 Å2), F167 (18.5 Å2), M171 (15.7 Å2), F311 (32.7 Å2), and F318 (19.4 Å2) had the highest contact surface area with resveratrol. To evaluate the impact of single-alanine mutations on the ligand-binding energy (ΔΔG, kcal/mol), we constructed 21 in silico alanine variants, calculated the binding free energy with the KDEEP server [43], and compared these energies to the wildtype enzyme. Alanine variants with the highest ΔΔG among the evaluated binding residues corresponded to F167 (0.46 kcal/mol), M171 (0.28 kcal/mol), F311 (0.4 kcal/mol), and F318 (0.33 kcal/mol) (Figure 3). The ConSurf [44] server was used to calculate a conservation score based on the VvROMT model and the multiple sequence alignment previously constructed for the phylogenetic analysis. Most of the residues in contact with SAM (W154, F167, M171, and D262) were highly conserved (score ≥ 7) except for W258. We selected the residues F24, F311, and F318 as hotspots, considering their high BSA, the ligand-binding affinity change (ΔΔG), and low conservation score. We further included W20 based on its high BSA value and its possible structural role in the substrate-binding site formation.

We constructed, expressed, and purified alanine variants of each selected critical residue to understand their relevance in the VvROMT enzymatic activity and to validate our VvROMT model.

### 2.4. Evaluation of Critical Residues

We measured the conversion rate of the alanine variants W20A, F24A, F311A, and F318A by LC-MS/MS (Table 1). F24A was the most impaired variant, followed by W20A, F311A, and F318A. When reactions started with resveratrol, less than 3.2% of pinostilbene was detected in all variants after 16 h of reaction, and no pterostilbene production was detected for variants W20A, F24A, and F311, while a residual conversion (6.5%) was observed for F318A. In reactions using pinostilbene as a substrate, less than 50% was converted to pterostilbene by all variants, while the wildtype enzyme converted >99%. According to the VvROMT model, residues F24 and W20 are located in the dimerization domain and are likely required for the correct formation of the substrate-binding cavity. Due to their location, an alternative explanation for the reduced enzymatic activity is impaired dimerization. We evaluated both alanine variants by size exclusion chromatography (SEC) and found that they were present in a dimeric state (Appendix A).

### 2.5. Rational Design of Substrate-Selective VvROMT Variants

In order to identify binding pocket residues that conveyed the one-step monomethylation of stilbenes we employed different approaches. First, we generated a multiple sequence alignment (MSA) of type-I OMTs with different regioselectivities for stilbene hydroxyl groups (Appendix A), and subsequently compared their binding site residues with the predicted ones of our VvROMT structure model.

For the MSA we considered the promiscuous enzymes AtOMT1 and OsOMT1 with 3/5-OH regioselectivity, and also the enzymes SbOMT3, PsPMT2, and PsPMT1 with a 3-OH methylation preference. Second, we analyzed a recent work of Guan Yuekai and co-workers (2020) [29], who performed directed evolution of VvROMT by error-prone PCR, generating the variants F311W, F311W/V119L, F311Y/E330Q, F311K/N46Q, F311Y, F311W/S67T/G300A, F311K/C69Q/V118I, and F311Y/S9T, which produced more pinostilbene compared with the wildtype enzyme. All their variants had a F311 substitution. Interestingly, we identified F311 as a critical residue according to our in silico and experimental analysis (Figure 3, Table 1). In their work, the highest pinostilbene titer was achieved with the F311W variant (103.58 mg L^−1^), followed by F311W/S67T/G300A (98 mg L^−1^), F311W/V119L (93.17 mg L^−1^), and F311Y (77.44 mg L^−1^) compared with 24.2 mg L^−1^ of the wildtype enzyme. Interestingly, SbOMT3 also has a tyrosine at the equivalent position (Appendix A). Furthermore, Hong and co-workers (2009) [45] performed site-directed mutagenesis with OsOMT1. This enzyme di-methylates the flavonoid tricetin (5,7,3’,4’,5’-pentahydroxyflavone) to yield tricin (3’,5’-di-O-methyltricetin). As a result, they obtained the 3′-OH variant H328R which monomethylated tricetin and produced selgin (3’-O-methyltricetin) but with compromised catalytic activity. The equivalent position of H328 corresponds to F318 in VvROMT, one of our identified critical residues (Figure 3). Additionally, we evaluated the VvROMT gene family from *V. vinifera* var. PN 40024, characterized by Parage in her Ph.D. thesis [46]. She measured the enzymatic activity of six VvROMT enzymes from a total of 11 cloned genes. The enzyme ROMT13 showed an increase in the substrate scope acceptance after a multi-substrate enzymatic activity assay, but preferably monomethylated resveratrol and 3,5-dihydroxyanisole, yielding pinostilbene and 3,5-dimethoxyphenol, respectively. ROMT13 has 77.8% sequence identity with VvROMT and just four residues changed when both substrate-binding sites were compared: L117F, F311W, T314L, and F318V. Therefore, we included this enzyme in our analysis. Finally, we used RhOOMT4 as a control of a 3/5 regioselectivity enzyme, because its expression in grapevine cell culture generates more pterostilbene than VvROMT [31].

According to this information, we focused on two regions to perform the rational design to modify the substrate selectivity of VvROMT. The selected positions were L117 and the range of residues F311, T314, F318, and A319. These residues interact with resveratrol based on the structural model and are likely related to OMT-I substrate selectivity based on the analysis above (Figure 4). We constructed eight VvROMT variants by site-directed mutagenesis, and their product conversion was measured by HPLC-UV.

### 2.6. Enzymatic Activity of VvROMT Variants and Structural Analysis

The positions selected for the rational design allowed us to generate resveratrol-selective VvROMT variants. We measured their enzymatic activity after 24 h of reaction with HPLC-UV. The substitutions with the highest impact on substrate selectivity involved the residues L117F and F311W. Modifications at positions 314, 318, and 319 mainly decreased enzymatic activity.

The F311W variant, previously reported [29], maintained the wildtype’s total conversion rate from resveratrol of approximately 44%, of which 13% corresponded to pinostilbene. On the other hand, L117F reached a total conversion of 71% from resveratrol with a 50% conversion to pinostilbene (Figure 5A). Interestingly, a synergistic effect was observed in the F311W/L117F variant, which had the highest conversion (74%) with a strong preference for monomethylating resveratrol, yielding pinostilbene as the principal product (67%).

Although the wildtype enzyme accepted both resveratrol and pinostilbene, we observed a preference to monomethylate pinostilbene, with a three times higher specific activity (32.5 nmol min^−1^ mg^−1^) than with resveratrol (11.1 nmol min^−1^ mg^−1^) (Table 2). When the reaction was started with resveratrol, no or only small amounts of pinostilbene were detected after 24 h of reaction. This suggests that all pinostilbene generated is rapidly methylated to pterostilbene, which is consistent with the work of Schmidlin and co-workers (2008), where pinostilbene, as an intermediary of the sequential two-step methylation reaction, appeared during short incubation times, while in more extended incubation periods, it was not detected [28].

Surprisingly, from resveratrol, the variant L117F had a more than two-fold increase in specific activity (26.6 nmol min^−1^ mg^−1^) compared to the wildtype, and a more than 1.5-fold increase when the reaction started with pinostilbene (52 nmol min^−1^mg^−1^).

In the variant F311W, an inverted substrate preference was observed (Figure 5A, Table 2), with a 35-fold decrease in the specific activity with pinostilbene (0.91 nmol min^−1^mg^−1^) compared with the wildtype enzyme. The substrate preference inversion was even more evident in the F311W/L117F variant, where its activity for pinostilbene considerably decreased, and it was not possible to detect the di-methylated product after 1 hour of reaction, consistent with only a 19% conversion to pterostilbene after 24 h (Figure 5B).

Structurally, we observed that the L117F mutation increased the contact surface area of this residue by 78% and enabled edge-to-face aromatic interaction with the substrate’s B ring. This mutation likely increased the overall binding affinity, as seen in the L117F and the L117F/F311W variants when resveratrol was used as a substrate (Figure 5A). The F311W mutation further increased the contact surface area by 11% (aromatic stacking interaction with the B ring). Interestingly, this position is connected to F318 via a 17-residue α-helix (302–318), where F318 is in close contact with the 3-methoxy group. Thus, F311W could have a mediated effect on substrate selectivity, but further analysis is needed to understand this contribution.

In line with our rational design, another possible way to generate resveratrol-selective variants was to modify the residues within close vicinity of the 3-methoxy group of pinostilbene, preventing its binding. The selected positions were 314, 318, and 319.

The variant F318Y/A319N maintained 3/5-OH regioselectivity but had more than a two-fold decreased total conversion from resveratrol compared with the wildtype (Figure 5A). According to the VvROMT model, F318 stabilizes the stilbene’s A ring with aromatic stacking interactions. The substitution by another hydrophobic residue (tyrosine) may not affect the enzymatic activity. Therefore, this effect was likely influenced by the polar A319N substitution.

On the other hand, the substrate-selective variant F318R/A319N introduced a bulky and polar residue at the 3-methoxy binding pocket, which was likely detrimental to pinostilbene binding. However, the introduction of two polar residues in the generally hydrophobic substrate-binding site may substantially impact the enzymatic activity, consistent with the observation of the equivalent variant H328R in the OsOMT1 enzyme [45]. For the resveratrol-selective variants L117F/F311W/T314L/F318V and L117F/F311L/T314L-/F318L, the substitutions F318V and F318L likely disfavor binding of pinostilbene’s 3-methoxy group due to the bulky, non-planar side chains of valine and leucine, which explains the less than 5% pterostilbene conversion from pinostilbene (Figure 5B). However, a decrease in their enzymatic activity was also observed from resveratrol, which could be related to the T314L substitution. The position 314 is juxtaposed to the stilbene ethylene group and next to the 3-methoxy binding pocket. Mutation of the larger leucine residue confines the substrate-binding site volume close to the 3-methoxy binding pocket.

Through rational design we could generate variants highly selective for the one-step monomethylation reaction of resveratrol. In contrast, wildtype VvROMT preferentially di-methylates resveratrol to pterostilbene in a two-step reaction.

## 3. Materials and Methods

### 3.1. Experimental Section

#### 3.1.1. Chemical Reagents

Trans-resveratrol 99%, pinostilbene 97%, and pterostilbene 99% were purchased from AK Scientifics Inc. CA, USA; S-adenosyl methionine (SAM) p-toluenesulfonate salt from AstaTech Inc. Bristol, PA, USA or AK Scientifics Inc., Union City, CA, USA; ethyl acetate from Merck; and formic acid 88% and acetonitrile HPLC grade from Bioslabchile Ltda. Chemically competent cells Top10 and BL21(DE3) were purchased from ThermoFisher Scientific.

#### 3.1.2. Gene Synthesis and Cloning

An *E. coli* codon-optimized sequence of resveratrol O-methyltransferase, accession no. FM178870 (GenBank), from *Vitis vinifera* was purchased from GenScript (Piscataway, NJ, USA) Appendix A. By digestion with NcoI and XhoI (NEB), the VvROMT coding sequence was cloned into the pET25-GB1 expression vector (kindly donated by Professor César Ramirez, UC). The final construct contained a histidine-tagged GB1 protein to increase VvROMT solubility, separated by a TEV protease cleavage site. The pET25-GB1-ROMT plasmid was transformed into chemo-competent *E. coli* strain Top10 and confirmed by sequencing (Macrogen Inc., Seoul, Korea).

#### 3.1.3. Site-Directed Mutagenesis

Different positions predicted by the in silico analysis were used to generate VvROMT variants. W20A, F24A, F311A, F318A, and L117F/F318L were constructed by Gibson Assembly protocol [47], using pET25-GB1-ROMT or L117F as a template for the amplification procedure. Primers list sequences are shown in the supporting information (Appendix A). PCR was performed under the following conditions: 98 °C for 3 min, 35 cycles of amplification at 98 °C for 15 s, 63 °C for 30 s, and 72 °C for 3 min, and final extension at 72 °C for 7 min. Fragments were purified using the Wizard^®^ purification Kit (Promega, Madison, WI, USA). Then, a proper amount of each DNA fragment was used for the Gibson Assembly procedure (according to the length-ratio described in the protocol [48]). Final constructs were confirmed by sequencing in Macrogen Inc., Seoul, Korea.

All the other variants were generated by the QuickChange strategy (Agilent Technologies, Santa Clara, CA, USA). Primer list is shown in the supporting information (Appendix A). Temperature program: 98 °C for 30 s, 30 cycles of amplification at 98 °C for 15 s, 62–65 °C for 20 s, and 72 °C for 4 min, and a final extension at 72 °C for 5 min. Amplification products were digested with DpnI (NEB) (37 °C, 2 h) and used directly to transform chemo-competent *E. coli* strain Top10. Final constructs sequence was confirmed by Microsynth AG, Balgach, Switzerland. All variant plasmids were transformed into *E. coli* strain BL21 (DE3) for protein expression.

#### 3.1.4. Expression and Protein Purification

An *E. coli* strain BL21 (DE3) culture, containing pET25-GB1-ROMT vector, was grown in Terrific Broth (1.2% tryptone, 2.4% yeast extract, 0.3% glycerol, and 100 mM phosphate buffer, pH 7.4) supplemented with kanamycin at 50 µg mL^−1^. Then, cells were cultivated at 37 °C at 150 rpm until OD_600_ 0.6–0.8 was reached. VvROMT expression was induced by adding IPTG at a final concentration of 0.5 mM. After 17 h of incubation at 28 °C and 150 rpm, cells were harvested by centrifugation (4000 rpm, 4 °C for 30 min) and resuspended in buffer A (20 mM Tris-HCl, pH 7.5, 500 mM NaCl, 10% glycerol, 20 mM imidazole) plus 10 mM DTT, 1 mM PMSF, and 10 U DNAase. Lysis was performed by sonication for 7 min (7 seg ON 15 seg OFF) on ice, with a Q125 Sonicator (QSonica, Newton, CT, USA). Lysate was centrifuged (13,000 rpm, 30 min at 4 °C), and the clarified supernatant was passed through a sterile 0.22 µM MCE membrane BIOFIL syringe filter. Soluble VvROMT expression was confirmed by SDS-PAGE (Appendix A).

VvROMT purification was carried out through immobilized metal affinity chromatography (IMAC), with HisTrap FF column (GE Healthcare Biosciences, Pittsburgh, PA, USA) according to the manufacturer’s instructions. The clarified extract was loaded onto the column, which was preconditioned with buffer A, and after 10 bed column washes, the fusion protein was eluted with elution buffer (20 mM Tris-HCl, pH 7.5, 500 mM NaCl, 10% glycerol) using a four-step imidazole gradient (50, 100, 180, and 250 mM) or a linear gradient. The selected fractions were pooled according to visualization by SDS-PAGE or by absorbance at 280 nm. Buffer-exchange using Amicon Ultra-15 Centrifugal Filter Units (NMWL 30 kDa; Merck Millipore) was performed twice with buffer A, without imidazole. GB1-ROMT fusion protein was subjected to proteolytic removal of the N-terminal GB1 protein using a His-tagged TEV protease followed by one-step final purification with IMAC system, as mentioned above.

Protein concentration was determined with Bradford assay (Biorad, Hercules, CA, USA) using BSA as standard. Purification of ROMT variants was conducted in the same manner. Proteins were used immediately for enzymatic reactions or stored at −80 °C with 20% of glycerol.

#### 3.1.5. Enzymatic Assay

The enzymatic reactions were performed according to Schmidlin, L. et al., (2008) [28] with some modifications. Briefly, the reaction mix contained 1.25 µM (0.05 µg/µL) of the purified wildtype enzyme or variants, 400 µM of SAM, 62.5 µM of the substrate prepared in DMSO, 100 mM Tris-HCl, 5 mM MgCl2, and 0.5 mM DTT, pH 7.5, in a final volume of 200 µL at 30 °C. Phenolic compounds were extracted with 200 μL of ethyl acetate, vortex for 30 s and centrifuged for 5 min at 6000 rpm. Extracts were evaporated to dryness at 50 °C and then dissolved in 330 µL of 80 % (*v*/*v*) acetonitrile. These samples were used directly or diluted for LC-MS/MS analysis. The enzymatic activity (% conversion) was determined relative to the total concentration of substrate and products measured after 16 h of reaction.

The reaction was adapted for HPLC-UV, using a reaction mix containing 5 µM (0.2 µg/µL) of the purified enzyme, 700 µM of SAM, 350 µM of the substrate prepared in DMSO, 100 mM Tris-HCl, 5 mM MgCl2, and 0.5 mM DTT, pH 7.5, in a final volume of 600 µL at 30 °C and 600 rpm. Then, 100 µL of each sample was quenched using 150 μL of acetonitrile (containing 400 mM HCl) followed by centrifugation. Samples were used directly for HPLC-UV analysis. The enzymatic activity (% conversion) was determined relative to the total concentration of substrate and products after 24 h. The specific activity was measured when less than 10% of product conversion was reached. From resveratrol, the specific activity was calculated by adding the specific activities of both products, pinostilbene and pterostilbene (pterostilbene concentration was multiplied by 2, considering this product previously was pinostilbene in the two-step methylation reaction).

All the reactions were performed in darkness to protect from stilbene trans/cis isomerization.

#### 3.1.6. High-Performance Liquid Chromatography

##### LC-MS/MS

To separate and quantify products of alanine VvROMT variants reactions, we used an Eksigent Ekspert Ultra LC100-XL (AB/Sciex, Concord, ON, Canada) coupled with a TripleQuadTM 4500 mass spectrometer system. The chromatographic separation was carried out using the column Inersil ODS-4 (3 μm, 2.1 × 100 mm) (GL Sciences) at 40 °C and a gradient elution of 0.1% formic acid in water (A) and acetonitrile (B) as a mobile phase. The gradient was programmed as follows: 0–2 min, 30% B; 2–4 min, 50% B; 4–5.5 min, 70% B; and 5.5–6.5 min, 30% B. The injection volume was 10 µL, and the flow rate was kept at 0.5 mL/min.

For MS/MS analysis, detection conditions were the following: electrospray ionization (ESI) in negative mode and a multiple reaction monitoring (MRM) mode were selected. The optimized parameters applied to each compound are presented in (Appendix A). The quantification of substrate and products was carried out according to a calibration curve with the standards resveratrol, pinostilbene, and pterostilbene, in a concentration range between 0.05 to 3.5 mg L^−1^ (Appendix A) dissolved directly in 80 % (*v*/*v*) of acetonitrile. An example of the transition chromatogram is shown in Appendix A. Samples were prepared in quadruplicates.

##### HPLC-UV

Conversion rate of VvROMT variants was measured by HPLC (Appendix A) on an Agilent Technologies 1100 Series with DAD detector and autosampler. The chromatographic separation was carried out at 30 °C using the reversed-phase Nucleodur C_18_ Pyramid column (250 × 4.6 mm, 5 μm, Macherey Nagel).

A gradient elution of filtrated water (A) and acetonitrile (B) was used as a mobile phase. The gradient was programmed as follows: 0–12 min, 55% B; 12–16 min, 80% B; and 16–20 min, 55% B. Flow rate was kept at 0.5 mL/min and 2 µL of analyte was injected. Absorption was detected simultaneously at 254, 280, 306, and 320 nm. 306 nm wavelength was chosen for calibration and evaluation of resveratrol, pinostilbene, and pterostilbene compounds. For calibration, a 10 mM stock of substrate in DMSO was used to prepare dilutions in reaction buffer (100 mM Tris-HCl, 5 mM MgCl2, pH 7.5) in a range of 10–500 µM. 100 µL of each sample was diluted with 150 μL of acetonitrile (containing 400 mM HCl). All calibration and samples were prepared at least in duplicate.

Calibration curve equation for resveratrol (1), pinostilbene (2), and pterostilbene (3)
(1)$$Y=2.63X-36.08;R^{2}=0.985$$
(2)$$Y=1.53X+34.39;R^{2}=0.986$$
(3)$$Y=1.32X-5.39;R^{2}=0.996$$

#### 3.1.7. Size Exclusion Chromatography

SEC experiments were performed on the Äkta Purifier LC System (GE Healthcare, Stockholm, Sweden) using the column SuperdeX 75 10/300 GL (GE Healthcare). The column was equilibrated with buffer 50 mM Tris-HCl, pH 7.5, using a flow rate of 0.5 mL/min. For column calibration, LMW Gel Filtration kit (GE Healthcare) at 3 µg/µL in 50 mM Tris-HCl, pH 7.5, 100 mM NaCl buffer was used. Calibration proteins were albumin (67 kDa), ovalbumin (45 kDa), chymotrypsinogen A (25 kDa), and ribonuclease A (18.7 kDa) (Appendix A). Molecular weight of alanine variants W20A and F24A, compared with the wildtype VvROMT enzyme (Appendix A), was evaluated at 2 µg/µL in a range of 150 to 300 µL.

### 3.2. Computational Methods

#### 3.2.1. Phylogenetic Analysis

We employed BLASTp to search for VvROMT related OMT-I sequences in the UniProt/Swiss-Prot database. Two cutoffs were applied for the resulting sequences: over 30% of sequence identity and an e-value of 8 × 10^−30^. Furthermore, we included by literature enzymes that O-methylated stilbenes and were not present in the above-mentioned sequences. A total of 63 final OMT-I sequences were aligned with PROMAL 3D [48] including six OMT-I protein structures, (PDB IDs: 1FP2, 1KYZ, 2QYO, 3P9i, 5ICE and 3REO) and the enzyme PlOMT, demethylpuromycin-OMT (UniProt code: V9W3E0), which was used as an outgroup. The phylogram was inferred by using the Maximum Likelihood (ML) method with MEGA X software [49], and their reliability was assessed by bootstrapping (1000 iterations). The tree with the highest log likelihood (−20189.7) was selected. The tree was drawn to scale, with branch lengths measured as the number of substitutions per site and was visualized with Evolview server [50].

#### 3.2.2. Homology Modeling and Molecular Docking

To build the three-dimensional protein model of VvROMT, we used the TopSearch server [51] to identify an appropriate template, based upon the structure of the isoflavone 4′-OMT (PDB ID:1ZG3). The criteria for template selection included OMT-I, amino acid sequence identity, crystal resolution, Rfree, presence of co-crystallized ligands, and a closed conformation. Two templates were selected, isoflavone OMT from *M. sativa* (PDB ID: 1FP2) and the caffeic acid OMT, LpCOMT1 from *L. perenne* (PDB ID: 3P9I), with 48.2% and 33.1% seq. identity with VvROMT, respectively. A dimeric model of VvROMT in a catalytically competent closed conformation was generated using MODELLER, version 9.16 [52], with a multiple template procedure. Sequences were aligned using Clustal Omega [53] and merged manually to yield the final dimeric alignment. The most accurate model among 20 was chosen based on the normalized DOPE score [54] and SAVES server [55].

VvROMT model energy minimization was carried out with MOE [56] using the AMBER12 force field before the docking procedure. Then, a list of five molecules, including two controls, was used to evaluate the binding pocket. The active site was defined at a 14 Å radius around the CZ atom of F167, using the GOLD suite [57] 5.1 (Cambridge crystallographic Data Centre), and default set parameters were selected. Then, a total of 50 independent genetic algorithm (GA) runs were performed per molecule. Finally, the GoldScore scoring function was used to evaluate each ligand’s binding position, and the average of the top ten docking scores was selected.

#### 3.2.3. Critical Residues Identification

Binding residues were defined as all residues within 6 Å from resveratrol in our VvROMT model. A total of 21 residues were included. In silico alanine variants of these residues were generated with MODELLER software and the ligand-binding affinity (ΔΔG, kcal/mol) was estimated with the KDEEP server [43]. We further evaluated the residue conservation pattern with the ConSurf server [44] based on the multiple sequence alignment previously generated by PROMAL 3D [48]. The ConSurf score was represented by an increasing conservation scale from 1 to 9. The buried surface area (BSA) was calculated for the binding pocket residues with our dr_sasa software tool [42].

## 4. Conclusions

We successfully generated a three-dimensional VvROMT model in a closed and catalytically competent conformation in complex with SAM and resveratrol, which allowed us to implement a rational design strategy to modify the substrate preference of VvROMT. Combining the information obtained from different in silico tools and published data with an in-depth structure and sequence analysis of the binding sites of others, stilbene OMT-I enabled us to generate resveratrol-selective variants. These enzymes generated pinostilbene as the principal product. The variant L117F presented an overall improvement of the enzymatic activity and F311W/L117F was the most resveratrol-selective variant, producing mostly pinostilbene after 24 h of reaction. Both enzymes showed 1.5-fold increased total conversion compared to the wildtype enzyme. These variants could be recombinantly expressed in both metabolically engineered plant cell cultures and microbial systems to efficiently produce the nutraceutical pinostilbene. Indeed, we propose that VvROMT could be tailor-made to be applied to diversify production of stilbenes or other related O-methylated phenolic compounds.

## Data Availability

Data is contained within the article or Appendix A. The three-dimensional VvROMT model is available at https://dx.doi.org/10.5452/ma-ruiiq (accessed on 13 April 2021).

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
