# Peer review of "Rational Design of Resveratrol O-methyltransferase for the Production of Pinostilbene"

_ijms, 2021, doi:10.3390/ijms22094345_

Round 1
Reviewer 1 Report
In this paper, the authors present the rational design of O-methyltransferase for the production of pinostilbene. Since crystal structure is not available for the enzyme of their choice, the group constructed a three-dimensional model and identified important binding site residues by applying various in silico approaches. Substituting these residues to alanine resulted in reduced enzymatic activity, validating the in silico model. The group also prepared protein variants using rational design and they were able to modify the substrate selectivity of the wild type enzyme, and also showed increase in enzymatic activity.
The paper is well structured and clearly written and the necessary control experiments were presented. Therefore, the paper can be published in the presented form.
Author Response
We sincerely appreciate the effort and comments of Reviewer 1.
Reviewer 2 Report
Resveratrol was a hot compound some 10 to 15 years ago, but the attractivity has considerably dropped. One reason is the poor water solubility of resveratrol and the poor bioavailability due to its rapid metabolism. Other groups proposed glycosylation or resveratrol to solve these problems. This group suggest the di-methylation of resveratrol. For this purpose, in silico approaches for substrate preference of VvROMT and production in E. coli strain BL21 (DE3) are described. The paper merits publication, but I would like to mention a few ideas for improvement also for future papers by this gourp. (1) A short comparative consideration would have been useful. For example, what kind of manufacturing process would the group finally propose for the resveratrol and subsequent pterostilbene production, as individual process steps can be chosen and combined from different procedures (chemical synthesis, plant extraction, enzyme cascade or whole cell, microbial or yeast recombinant, plant cell culture). DSM developed a chemical resveratrol process, but the recombinant fermentation processes by e.g. Fluxome or Evolva in Saccharomyces cerevisiae or Du Pont in oleaginous microorganisms are the most promising ones. Thus, the next would be to try to use their improved O-methyltransferase in a Evolva, Fluxome or other strain for the production of pterostilbene instead of resveratrol. (2) The protein production with E. coli containing pET25-GB1-ROMT vector in terrific broth supplemented with kanamycin, expression induced by IPTG (intracellular expression), purification by metal affinity chromatography is very far from a commercially viable process. For topics like these, which are not dealing with basic research but dealing with a marketed product and existing industrial processes, I would expect that the authors at least try to put their findings into a real-world context – this for another similar publication you might plan. (3) Line 104, 564 “variants could be expressed in different host cell platform”: what is meant by different host cell platforms? This has to be specified.
Author Response
We sincerely appreciate the detailed revision of our manuscript. Points (1) and (2) are helpful suggestions for future publications that we will gladly consider.
With respect to the last point “(3) Line 104, 564 “variants could be expressed in different host cell platform”: what is meant by different host cell platforms? This has to be specified.” we modified lines 58-66 to clearly describe the different strategies to produce stilbenes, highlighting bioproduction as a more efficient and environment-friendly process. We further included possible host options in lines 107 and 571 of the manuscript to integrate our VvROMT variants into a biosynthetic pathway to produce pinostilbene. Among them, the most interesting to us are the de novo biosynthesis of resveratrol and pterostilbene from glucose in E. coli and also in S. cerevisiae developed by Li et al. (2016) and Kyung Taek et al. (2017), respectively. These strategies not only integrate a complete synthetic pathway in the host of choice but also its metabolic optimization. On the other hand, the use of grapevine cellular cultures seems competitive (Martínez-Márquez et al., 2018). Integration of our specific enzymatic variants seems straightforward, since these cells naturally produce resveratrol, and its induction can be regulated.
Additionally, we performed minor modifications of typographical errors and rephrasing to improve the overall clarity at the lines: 84, 189, 191, 219, 340, 396, 417, 432, 441, 463.
We hope that these modifications resolve the issues addressed by Reviewer 2.
References
Li, M.; Schneider, K.; Kristensen, M.; Borodina, I.; Nielsen, J. Engineering Yeast for High-Level Production of Stilbenoid Antioxidants. Sci. Rep. 2016, 6, 36827. https://doi.org/10.1038/srep36827.
Kyung Taek, H.; Sun-Young, K.; Young-Soo, H. De Novo Biosynthesis of Pterostilbene in an Escherichia Coli Strain Us-ing a New Resveratrol O-Methyltransferase from Arabidopsis. Microb. Cell Fact. 2017, 16, 30–38. https://doi.org/10.1007/s00253-009-1975-y.
Martínez-Márquez, A.; Morante-Carriel, J. A.; Palazon, J.; Bru-Martínez, R. Rosa Hybrida Orcinol O-Methyl Transferase-Mediated Production of Pterostilbene in Metabolically Engineered Grapevine Cell Cultures. N. Biotechnol. 2018, 42 (December 2017), 62–70. https://doi.org/10.1016/j.nbt.2018.02.011.